# Synthesis of Novel Podophyllotoxin–Benzothiazole Congeners and Their Biological Evaluation as Anticancer Agents

**DOI:** 10.3390/ijms26136033

**Published:** 2025-06-24

**Authors:** Pramukti Nawar Rai’dah, Zuzanna Molęda, Aleksandra Osińska, Armand Budzianowski, Izabela Młynarczuk-Biały, Zbigniew Czarnocki

**Affiliations:** 1Faculty of Chemistry, University of Warsaw, Pasteura 1, 02-093 Warsaw, Poland; p.raidah@uw.edu.pl (P.N.R.); moleda@chem.uw.edu.pl (Z.M.); a.osinska12@student.uw.edu.pl (A.O.); 2National Centre for Nuclear Research, A. Sołtana 7, 05-400 Otwock, Poland; armand.budzianowski@ncbj.gov.pl; 3Department of Histology and Embryology, Medical University of Warsaw, Chałubińskiego 5, 02-004 Warsaw, Poland

**Keywords:** podophyllotoxin, benzothiazole, anticancer, cytotoxicity, tubulin

## Abstract

A series of novel podophyllotoxin derivatives containing benzothiazole scaffolds were synthesized and evaluated for their in vitro cytotoxic activity against five cancer cell lines (MCF-7, SKOV-3, B16F10, LOVO, and HeLa). Two compounds, **7** and **11**, which are different only by the absence or presence of the ester group, showed the strongest cytotoxic effect towards all tested cancer cell lines with the IC_50_ 0.68–2.88 µM. In addition, it was demonstrated that these compounds inhibit cancer cell proliferation by inducing G2/M phase arrest in HeLa cells. The structure–activity relationship was analyzed and it confirmed the importance of the core structural features like a dioxolane ring and free-rotating trimethoxyphenyl group for cytotoxicity. Moreover, the *R* configuration of the ester group at the C-8′ position proved to be substantial since its epimer was inactive. The molecular docking studies revealed that the most potent compounds have a different binding mode to β-tubulin than podophyllotoxin; however, the benzothiazole fragment docked in a similar location as the trimethoxyphenyl group of podophyllotoxin, exhibiting similar hydrophobic interactions. These findings clearly indicate that podophyllotoxin–benzothiazole derivatives could be addressed for further pharmacological studies in anticancer research.

## 1. Introduction

Podophyllotoxin **1** is an active compound from *Podophyllum* sp. that is well known for its broad spectrum of biological activities such as anticancer, antiviral, anti-inflammatory, and antibacterial properties [1,2,3,4]. Molecular mechanisms of podophyllotoxin **1** activity involve the inhibition of tubulin polymerization into microtubules and blocking the replication of cellular DNA; thus, it is considered an antitumor agent [4,5]. Unfortunately, despite such a promising bioactivity, podophyllotoxin **1** is highly toxic against normal cells; therefore, it is used only for the topical treatment of anogenital warts [6].

Cancer remains one of the leading causes of global mortality, with 20 million new cases and 9.7 million deaths reported in 2022 [7]. Breast and colorectal cancers are the second and third most prevalent malignancies, respectively, while cervical and ovarian cancers pose a significant concern due to high mortality rates among women [7,8]. Additionally, the incidence of melanoma continues to rise, correlating with increased exposure to ultraviolet (UV) radiation [9]. Numerous strategies have been developed to find novel drug candidates with high cytotoxicity to cancer cells, sufficient bioavailability, and acceptable toxicity to normal cells. Up to now, many podophyllotoxin **1** derivatives have been reported. The modifications in the C-ring of podophyllotoxin were predominantly explored. Among the most promising derivatives, etoposide **2** and teniposide **3** (Figure 1) were introduced in anticancer therapy [10]. Interestingly, these two derivatives have a different molecular mechanism of activity compared to the parent compound. While podophyllotoxin **1** inhibits the assembly of tubulin into microtubules, etoposide **2** and its analogs inhibit DNA topoisomerase, preventing relegation of the double-strand break [11]. The core structural features of podophyllotoxin **1**, like *trans*-fused C-lactone, a fused dioxolane ring, and the almost-orthogonal free-rotating 3,4,5-trimethoxyphenyl fragment, are considered essential for the antitubulin activity of podophyllotoxin derivatives [12,13,14]. Although many of them have been synthesized so far, the most explored strategy is based on modifications at the C-7 position [10]. However, a literature survey revealed that podophyllic aldehyde **6** and its analogs lacking a *trans*-lactone D-ring exhibit significant cytotoxicity against various tumor cell lines [15,16]. Therefore, modification of the D-ring of podophyllotoxin **1** became of interest.

KL-3 **7** is a derivative of podophyllotoxin **1** (Figure 1) that was developed in our team [17,18]. It contains a benzothiazole group instead of the lactone ring. The synthesis of KL-3 **7** can be effectively performed by the transformation of podophyllic aldehyde **6**, the preparation of which was based on the photochemical cyclization [19]. In contrast to podophyllotoxin **1**, which is highly toxic, KL-3 **7** allows for the repair of normal cells and triggers mechanisms that restore non-cancer cell homeostasis [18]. This promising result inspired us to design and study new KL-3 **7** analogs to obtain more potent drug candidates and evaluate their structure–activity relationships. We planned three modifications on **7**, including adding substituents at the benzothiazole ring, modification of the ester part with hydrazines, and demethylation at the ring E. However, despite our efforts, the ester group modifications brought unexpected results. Nevertheless, we confirmed the leading structure for this class of podophyllotoxin–benzothiazole congeners with potent anticancer activity.

## 2. Results and Discussion

### 2.1. Chemistry

Figure 1 illustrates the synthetic pathway for compounds **7**–**9**. The unsaturated aldehyde **6** was prepared according to the procedure previously described [15]. In short, the hydroxyl group in podophyllotoxin **1** was oxidized with pyridinium dichromate (PDC) and the resulting product was hydrolyzed in acidic conditions, using sulfuric acid in methanol to open the lactone ring. This strategy enabled us to maintain the desired *cis*-stereochemistry of substituents at C-7′ and C-8′ positions, while hydrolysis under alkaline conditions causes undesired epimerization at C-7′ position [12,20]. We observed that careful control of the reaction time enables us to obtain a good yield of compound **4**, while extending this time beyond one hour resulted in the formation of compound **5**. This result was confirmed with NMR by the appearance of additional methyl protons at 3.25 ppm and methoxy carbon at 70.2 ppm. The reduction of **4** with sodium borohydride afforded 1,3-diol, which was transformed into unsaturated aldehyde **6** using Swern oxidation. The ^1^H and ^13^C NMR of compounds **4** and **6** were identical, as reported in the literature [15]. Compounds **7**–**9** were synthesized by the condensation of aldehyde **6** with corresponding 2-aminothiophenol derivatives in the presence of sodium metabisulphite as an oxidizing agent [21].

As shown in Figure 2, we further modified compound **7**, obtaining six derivatives (**10**–**15**). Attempted hydrazinolysis of compound **7** gave rise to compound **10** as a result of epimerization of the ester group, as confirmed by the ^1^H NMR spectrum, in which the methyl ester protons were shifted to 3.63 ppm. Alternatively, compound **11** was obtained by treatment of **7** with potassium hydroxide in methanol, followed by neutralization with hydrochloric acid. The absence of a ^1^H NMR signal of methyl protons from the ester group, and the appearance of two new methylene protons at 3.38 and 3.18 ppm, confirmed the unexpected removal of the ester group. On the other hand, the one-pot reaction of **7** with potassium hydroxide solution followed by the reaction with oxalyl chloride and hydrazine treatment led to aromatization of ring C **12**, which was identified from the disappearance of two methine protons in the range 4.0–5.0 ppm and the appearance of two quaternary carbons at 129.2 ppm and 132.9 ppm. The reaction of **7** with DIBAL-H resulted in the formation of a complex mixture of products difficult to identify, from which only compound **13** was isolated**.** Considering the fact that many podophyllotoxin derivatives with a free 4′-hydroxy group, such as etoposide **2**, have potential as anticancer agents, especially targeting topoisomerase II [22,23,24], we considered performing a demethylation of **7**. Thus, the reaction of **7** with methionine in the presence of methanesulfonic acid was performed according to the procedure previously reported [23]; however, we observed that not only did demethylation occurred, but the dioxolane ring was also opened to form compound **14**. Fortunately, the reaction of compound **7** with trimethylsilyl iodide (TMSI) [25] successfully produced compound **15**.

The absolute configurations of compounds **7**, **9**, and **10** were assigned by X-ray diffraction analysis. Compound **7** and its derivative **9** crystallize in the orthorhombic *P*2_1_2_1_2_1_ space group with Z = 8 molecules in the unit cell, whereas crystals of **10** crystallize in the monoclinic *P*2_1_ space group with Z = 4. Although crystals are built of the single diastereoisomer, which is consistent with the non-centrosymmetric space group, the independent part of the unit cell is composed of two molecules (Z’ = 2) in each crystal of the compounds **7**, **9**, **10**. Two molecules differ in geometrical parameters: bond lengths and angles. The most spectacular difference is, however, the difference in the conformation of the methoxy group of the phenyl ring. In general, molecules in the crystal structure are connected by a series of weak CH···O and CH···N hydrogen bonds, which are probably responsible for the conformational richness of the molecules in the solid state.

In all cases, the data were collected with full Friedel pair coverage, ensuring sufficient sensitivity to anomalous scattering effects to allow reliable determination of the absolute configuration. The resulting Flack parameter values, which are close to zero, confirm the accurate assignment of the enantiomeric form in each case and the high reliability of the absolute structure determination. The compound configuration was determined as (7′R,8′R) for compounds **7** and **9**, and (7′R,8′S) for compound **10**, as shown in Figure 2.

### 2.2. Biological Evaluation

#### 2.2.1. The In Vitro Cytotoxicity Assay

The cytotoxicity properties of semi-synthesized compounds **7**–**15** was evaluated against five types of cancer cell lines including, human: HeLa (cervix cancer), LoVo (colorectal cancer), MCF-7 (breast cancer), SKOV-3 (ovarian cancer), and murine: B16F10 (malignant melanoma). Thus, various human tumor cell lines representing major types of malignant cancers were included in the study protocol. Non-tumorigenic HaCaT cells were used as a representative model of normal (non-cancerous) cells. The crystal violet assay and PrestoBlue assay were used to evaluate the cytotoxic/cytostatic effect. All compounds showed diverse cytotoxicity activity against all the tested cell lines, as shown in Table 1. We also included the cytotoxicity data of podophyllotoxin and etoposide as reference values. As expected, compound **7** showed a significant growth inhibition to all tested cancer cell lines; however, it was also significantly toxic to HaCaT cells. Compound **8** with an additional CF_3_ group on the benzothiazole ring was inactive on all tested cell lines, while compound **9** with a Cl-substituent was selective to HeLa and MCF-7 with IC_50_ 1.81–2.02 µM, and had moderate activity on SKOV-3 and LOVO. In addition, compound **9** showed higher inhibition to cell lines derived from cancer than to the non-tumorigenic one. In contrast, compound **10** showed no cytotoxicity on all tested cell lines, which is consistent with the literature data on similar compounds that were found to be inactive on HepG2 (human liver cancer cell) [26], A549 (lung cancer), HT-29 (colorectal cancer), and Mel-28 (melanoma) [27]. Interestingly, compound **11** revealed a strong cytotoxic/cytostatic effect against all tested cell lines with IC_50_ 1.54–2.88 µM, despite the absence of the ester group. Moreover, the IC_50_ value of this compound showed no significant difference with the parent compound **7**. On the other hand, compounds **12** and **13** showed no effect on cancer cells nor non-cancer cells, indicating that the presence of the methyl ester group and the free-rotating three-methoxy phenyl ring are important. Furthermore, compound **14**, lacking the dioxolane ring, showed low activity on LOVO, MCF-7, SKOV-3, and HaCaT cells. Lastly, compound **15** was found to exhibit moderate cytotoxicity on all the cell lines tested, except for LoVo cells with an IC_50_ in range 2.49–3.76 µM, and lower activity to HaCaT. Despite this, the demethylation product **15** displayed inferior activity compared to its parent compound, but still shows potential for future studies.

Additionally, the selectivity index (SI) was calculated as the ratio of the IC_50_ values for HaCaT and HeLa cells (SI = IC_50_HaCaT/IC_50_HeLa), where a higher SI value indicates the greater selectivity of the compound towards cancer cells. In general, all active compounds showed selectivity against HeLa cells, except for compound **12**. Compounds **9** and **15** demonstrated the highest selectivity, with SI values of 7.31 and 3.05, respectively. Although compounds **5** and **11** showed lower selectivity, their SI values were still above 1.00, and they are therefore still considered to be potent anticancer agents [28].

#### 2.2.2. Cell Cycle Analysis

To determine the possible mechanisms of compounds **7** and **11** in inhibiting the proliferation of cancer cells, cell cycle analysis was performed with the flow cytometry method, as shown in Figure 3. Compound **7** was previously reported to inhibit proliferation of HaCaT cells via cell cycle arrest in G2/M [18]. As a reference, HeLa cells were treated with the indicated concentrations of compounds **7** and **11** for 24 h and 48 h, and the cell cycle phases distribution of treated cells was determined with the propidium iodide (PI) method, as previously described [18]. The effects of novel podophyllotoxin derivatives on cell cycle phase distribution were evaluated in cells exposed to these compounds for 24 h or 48 h. Three compounds were compared: podophyllotoxin **1**, KL-3 derivative **7**, and its analog lacking the ester group **11**. Each compound induced cell cycle arrest in the G2/M phase. Compounds **7** and **1** were the most potent, with approximately 90% of the cells arrested in the G2/M phase. In contrast, compound **11**, at a low concentration of 5 µM, induced only a minor block in the G2/M phase. However, at higher concentrations and after 48 h of exposure, the percentage of cells arrested in the G2/M phase increased to approximately 43%. These findings suggest that the structure and concentration of podophyllotoxin derivatives significantly influence their ability to block the G2/M transition in the cell cycle.

### 2.3. Molecular Docking Studies

Podophyllotoxin is known as a tubulin polymerization inhibitor for disrupting microtubule formation, which leads to cell cycle arrest in mitosis [29,30]. While podophyllotoxin binds to tubulin at the same site as colchicine, the trimethoxyphenyl group binds to the active site of tubulin via hydrophobic interactions. Therefore, to predict the binding interaction between the synthetized compounds **7**, **11**, and **15** with β-tubulin (PDB: 1SA1), we performed a molecular docking analysis via SwissDock with AutoDock Vina 1.2.5 algorithm. SwissDock is a freely accessible web tool for small molecules, allowing docking with automatized ligand preparation and targeting. Moreover, this program exhibits a good accuracy of docking predictions with rapid execution times [31,32].

As a result of using this program, we found that compounds **7**, **11**, and **15** are subject to conformation change at the podophyllotoxin binding site, with a binding affinity −7.795 kcal/mol, −7.227 kcal/mol, and −7.965 kcal/mol, respectively, compared to the podophyllotoxin binding affinity equal to −7.682 kcal/mol. Interestingly, all docked benzothiazole groups (Figure 4a) were overlapping with the 3′,4′,5′-trimethoxyphenyl fragment of podophyllotoxin, having similar hydrophobic interactions with ALA316, CYS241, LEU255, and ALA250 [14]. The surface mode (Figure 4b) displayed that benzothiazole groups were located inside the binding pocket of β-tubulin, proving the crucial role of this structure motif responsible for potent inhibition [33,34].

Despite the prediction that the unsubstituted benzothiazole moiety plays an important role in inhibiting cancer cells, other structural elements such as a dioxolane ring, free-rotating 3′,4′,5′-trimethoxyphenyl group, and *cis*-methyl ester group also contribute to the binding mode. Therefore, careful and thorough analysis of the molecular mechanism of these potent compounds (**7**, **11**, and **15**) should be conducted.

### 2.4. The Structure–Activity Relationships

Using compound **7**, we analyzed the influence of substituents at the benzothiazole ring, the modification in ring A and ring E, and the presence of different functional groups at the C-8′ position. The structure–activity relationships (SARs) are presented in Figure 5. The results reveal that the presence of larger substituents at the benzothiazole ring leads to lower cytotoxicity (–H > –Cl > –CF_3_); therefore, it is clear that the unsubstituted benzothiazole ring is preferred for the desired bioactivity. In addition, the stereochemistry of the ester group has a crucial effect on its ability to inhibit cancer cells proliferation as compound **10** with *S*-configuration at C-8′ atom showed no effect on any of the tested cell lines. Compound **11** presents an interesting case because despite lacking *cis*-stereochemistry at the C ring, it still demonstrates significant anticancer activity. The fact that compound **13** shows no activity against cancer cells suggests that the hydroxymethyl group likely has insufficient interaction with the binding pocket. The aromatization of ring C in compound **12** resulted in the loss the cytotoxic activity to all tested cells. Finally, compound **15** better and broader antiproliferation activity than **14**, which proves that the dioxolane ring is important for anticancer activity.

### 2.5. Prediction of ADMET Parameter with SwissADME

To predict the pharmacokinetic properties and drug-likeness of the novel compounds, the ADME parameter (for absorption, distribution, metabolism, and excretion) was analyzed via the SwissADME web tool (http://www.swissadme.ch/, accessed on 9 April 2025) [35]. The physicochemical properties of prospective compounds **7**, **11**, and **15** (Table 2) suggest their promising drug-likeness, particularly for compound **11** with zero violations of Lipinski’s rule. In addition, compound **11** is predicted to have high GI absorption, indicating a high probability of passive absorption through the gastrointestinal tract [35,36]. However, experimental studies are required to confirm these predictions.

## 3. Materials and Methods

### 3.1. Chemistry

Podophyllotoxin (**1**) was obtained from (Sigma-Aldrich, St. Louis, MO, USA). Other chemicals were also obtained from Sigma-Aldrich. Etoposide-Ebewe (Ebewe Pharma GmbH, Unterach am Attersee, Austria), a ready-to-use infusion solution at a concentration of 20 mg/mL, was used in the study. The ^1^H NMR and ^13^C NMR spectra were recorded on a Bruker AVANCE apparatus, operating at 500 MHz and 300 MHz (^1^HNMR), and 125 MHz and 75 MHz (^13^C NMR), using tetramethylsilane (TMS) as an internal standard, with chemical shifts given in ppm. High-resolution mass spectra were recorded on Quatro LC AMD 604 apparatus using TOF MS ES + method. Melting points were measured on a micro melting point Fisher-John apparatus without correction. Optical rotations were measured with an Autopol IV Rudolph Research Analytical apparatus. X-ray crystallographic data were collected on Bruker ApexII Ultra. Column chromatography was carried out on silica gel SilicaFlash P60 230–400 mesh ASTM (Dichrom GmbH, Haltern am See, Germany), with the indicated eluents. Thin-layer chromatography (TLC) analysis used precoated Si-gel plates (Merck Kieselgel 60 GF254, 0.25 mm) (art. 5554). Anhydrous sodium sulfate was used as a drying agent. The organic solvents (MeOH, n-hexane, CHCl_3_, CH_2_Cl_2_, EtOAc) used in column chromatography were of the standard grade for purification.

#### 3.1.1. General Method of Synthesis

Compounds **4**–**6** were synthesized in accordance with the procedure previously described [15]. In brief, podophyllotoxin **1** was subjected to oxidation, followed by hydrolysis to form compound **4.** The reduction of the ketone group and subsequent Swern oxidation were carried out to afford compound **6**.

##### Compound **4**

For synthesis of compound **4**, podophyllotoxin **1** (2.05 g, 4.94 mmol, 1 eq) was dissolved in 40 mL of dichloromethane and placed on the ice bath. PDC (2.09 g, 5.55 mmol, 1.1 eq) was added and the mixture was stirred at 0 °C for 3 h. The reaction mixture was quenched with distilled water and extracted with dichloromethane. The organic extracts were collected, dried over anhydrous Na_2_SO_4_, and evaporated under reduced pressure. The crude product was purified with column chromatography using *n*-hexane-EtOAc (7:3) as the eluent. The NMR of the product was identical to the literature^4^. The resulting ketone (1.6 g, 3.9 mmol, 1 eq) was dissolved in MeOH and placed in an ice bath. Concentrated H_2_SO_4_ (15.8 mmol, 4 eq) was added dropwise and then the reaction mixture was stirred under reflux for 1 h. The solvent was evaporated and quenched with sodium bicarbonate, followed by extraction with EtOAc. The organic layer was dried over anhydrous Na_2_SO_4_ and evaporated. The crude product was purified by column chromatography to afford compound **4** with 75% yield. Prolongation of the reaction time for more than 2 h caused the formation of compound **5** as the main product. Compound **4** was isolated as white amorphous solid. ^1^H NMR (300 MHz, CDCl_3_): δ (ppm) = 7.53 (s, 1H), 6.58 (s, 1H), 6.11 (s, 2H), 6.04 (dd, *J* = 1.2, 4.2 Hz, 2H), 4.59 (d, *J* = 5.4 Hz, 1H), 4.23 (ddd, *J* = 3.0, 6.6, 10.8 Hz, 1H), 3.80 (s, 3H), 3.73 (s, 3H), 3.73 (s, 3H), 3.66 (s, 3H), 3.63 (dd, *J* = 6.0, 12.0 Hz, 1H), 3.06 (ddd, *J* = 3.0, 4.8, 12.9 Hz, 1H), 2.55 (t, *J* = 6.6 Hz, 1H). ^13^C NMR (75 MHz, CDCl_3_): δ (ppm) = 197.8, 171.6, 153.4, 153.4, 148.2, 140.9, 137.7, 134.2, 127.2, 108.6, 106.2, 106.0, 102.3, 61.9, 61.0, 56.3, 52.0, 47.3, 46.7, 45.4.

**Compound 5** was isolated as white powder. ^1^H NMR (300 MHz, CDCl_3_): δ (ppm) = 7.54 (s, 1H), 6.59 (s, 1H), 6.09 (s, 1H), 6.02 (dd, *J* = 1.2, 4.2 Hz, 2H), 4.58 (d, *J* = 5.1 Hz, 1H), 4.22 (dd, *J* = 2.4, 9.0 Hz, 1H), 3.80 (m, 1H), 3.79 (s, 3H), 3.72 (s, 6H), 3.68 (s, 3H), 3.54 (dd, *J* = 3.3, 9.3 Hz, 1H), 3.25 (s, 3H), 2.96 (dt, *J* = 3.0, 12 Hz, 1H). ^13^C NMR (75 MHz, CDCl_3_): δ (ppm) = 195.5, 171.9, 153.3, 153.0, 148.1, 140.3, 137.5, 134.4, 127.7, 108.4, 106.4, 106.0, 102.1, 70.4, 61.0, 56.5, 56.2, 51.9, 47.2, 46.1, 44.4. MS (EI) (*m*/*z*) for C_24_H_26_O_9_ [M + Na]^+^ = calc. 481.14735 found 481.14690.

##### Compound **6**

For the synthesis of compound **6**, initially, compound **4** was reacted with sodium borohydride, which afforded a 1,3-diol, and the NMR-data were in accordance with those previously described [15]. Next, the 1,3-diol was oxidized via Swern oxidation using oxalyl chloride, DMSO, and triethylamin. Compound **6** was purified by column chromatography with DCM-EtOAc (9:1–7:3) as the eluent and collected as yellow powder in 64% yield. The NMR data were identical to the literature [15]. ^1^H NMR (300 MHz, CDCl_3_): δ (ppm) = 9.60 (s, 1H), 7.36 (s, 1H), 7.27 (s, 2H), 6.89 (s, 1H), 6.62 (s, 1H), 6.46 (s, 1H), 6.00 (dd, *J* = 1.2, 4.6 Hz, 1H), 4.41 (dd, *J* = 0.9, 8.1 Hz, 1H), 3.98 (d, *J* = 7.8 Hz, 1H), 3.87 (s, 3H), 3.81 (s, 3H), 3.81 (s, 3H), 3.40 (s, 3H).

##### General Procedure for the Synthesis of Compounds **7**–**9**

Compounds **7**–**9** were prepared in accordance with the procedure previously described [17,18]. To a round-bottomed flask, compound **4** (0.25 mmol, 1 eq), corresponding 2-aminothiophenol (0.28 mmol, 1.2 eq), Na_2_S_2_O_5_ (0.28 mmol, 1.2 eq), and 2.5 mL DMSO were added. The mixture was stirred at 120 °C for 1.5–2 h under argon atmosphere. The mixture was then cooled to room temperature and water was added, causing the formation of yellow precipitate. The precipitate was collected by filtration and dried, followed by purification using column chromatography with *n*-hexane-EtOAc (8:2) as eluent. The recrystallization of **7** and **9** was carried out with anti-solvent crystallization method, using chloroform with additional *n*-hexane.

**Compound 7** was obtained as light green needles at 67% yield. m.p.: 168–169 °C. ^1^H NMR (300 MHz, CDCl_3_): δ (ppm) = 7.95 (ddd, *J* = 0.6, 1.3, 8.0 Hz, 1H), 7.84 (ddd, *J* = 0.6, 1.4, 7.9 Hz, 1H), 7.44 (ddd, *J* = 1.5, 7.0, 8.3 Hz, 2H), 7.35 (ddd, *J* = 1.0, 6.9, 8.3 Hz, 1H), 6.87 (s, 1H), 6.63 (s, 1H), 6.56 (s, 2H), 5.98 (dd, *J* = 1.5, 6.6 Hz, 1H), 4.61 (dd, *J* = 1.2, 7.5 Hz, 1H), 4.43 (d, *J* = 7.5 Hz, 1H), 3.89 (s, 3H), 3.84 (s, 3H), 3.84 (s, 3H), 3.43 (s, 3H). ^13^C NMR (75 MHz, CDCl_3_): δ (ppm) = 171.8, 167.3, 153.9, 153.5, 153.0, 148.7, 146.7, 137.5, 135.0, 134.6, 133.6, 131.8, 128.7, 127.1, 126.3, 125.5, 123.3, 121.6, 121.5, 109.0, 108.9, 107.7, 106.7, 101.5, 61.1, 56.3, 51.9, 49.2, 48.7. MS (EI) (*m*/*z*) for C_29_H_25_NO_7_S [M + H]^+^ = calc. 532.14232, found 532.14245.

**Compound 8** was obtained as a light green powder. Yield 63%.^1^H NMR (300 MHz, CDCl_3_): δ (ppm) = 8.21 (t, *J* = 0.9 Hz, 1H), 7.94 (d, *J* = 8.4 Hz, 1H), 7.58 (dt, *J* = 0.6, 8.4 Hz, 1H), 7.45 (s, 1H), 6.88 (s, 1H), 6.65 (s, 1H), 6.55 (s, 2H), 5.99 (dd, *J* = 1.5, 6.3 Hz, 1H), 4.62 (dd, *J* = 0.9, 7.5 Hz, 1H), 4.42 (d, *J* = 7.8 Hz, 1H), 3.89 (s, 3H), 3.84 (s, 3H), 3.84 (s, 3H), 3.43 (s, 3H). ^13^C NMR (75 MHz, CDCl_3_): δ (ppm) = 171.6, 169.2, 153.4, 153.4, 149.0, 146.6, 134.7, 134.7, 132.0, 128.1, 126.7, 122.2, 108.9, 108.9, 106.6, 101.5, 60.9, 56.2, 51.8, 49.0, 48.5. MS (EI) (*m*/*z*) for C_30_H_24_NO_7_F_3_S [M + H^+^] = calc. 600.12983, found 600.13097.

**Compound 9** was obtained as light yellow needles. Yield 72%. m.p.: 182–183 °C. ^1^H NMR (300 MHz, CDCl_3_): δ (ppm) = 7.92 (dd, *J* = 0.6, 2.1 Hz, 1H), 7.74 (dd, *J* = 0.6, 8.7 Hz, 1H), 7.42 (s, 1H), 7.32 (dd, *J* = 1.8, 8.4 Hz, 1H), 6.86 (s, 1H), 6.63 (s, 1H), 6.55 (s, 2H), 5.98 (dd, *J* = 0.9, 6.3 Hz, 2H), 4.60 (dd, *J* = 0.9, 7.5 Hz, 1H), 4.39 (d, *J* = 7.5 Hz, 1H), 3.89 (s, 3H), 3.84 (s, 6H), 3.43 (s, 3H). ^13^C NMR (75 MHz, CDCl_3_): δ (ppm) = 171.8, 154.8, 153.5, 153.5, 148.9, 146.7, 137.6, 134.9, 134.3, 132.8, 132.3, 132.0, 128.4, 126.9, 125.9, 123.0, 122.2, 109.0, 109.0, 106.7, 106.7, 101.6, 61.1, 56.3, 56.3, 51.9, 49.2, 48.6. MS (EI) (*m*/*z*) for C_29_H_24_NO_7_SCl [M + H^+^] = calc. 566.10292, found 566.10179.

##### Compound **10**

To obtain compound **10**, compound **7** (30 mg, 0.056 mmol, 1 eq) and hydrazine hydrate (0.428 mmol, 6 eq) were refluxed in 2-propanol (3 mL) for 2 h. The product was purified by preparative TLC (thin-layer chromatography) with CHCl_3_-MeOH (97:3) as eluent. The product was collected as colorless needles with 78% yield. The crystallization was performed from chloroform with additional *n*-hexane. M.p.: 198–200 °C. ^1^H NMR (300 MHz, CDCl_3_): δ (ppm) = 7.92 (dd, *J* = 0.6, 8,0 Hz, 1H), 7.82 (dd, *J* = 0.6, 7.4 Hz, 1H), 7.42 *(*td, *J* = 1.2, 7.2 Hz, 1H), 7.40 (s, 1H), 7.33 (td, *J* = 1.5, 7.0 Hz, 1H), 6.86 (s, 1H), 6.73 (s, 1H), 6.38 (s, 2H), 5.99 (dd, *J* = 1.2, 3.9 Hz, 2H), 4.67 (d, *J* = 2.1 Hz, 1H), 4.61 (d, *J* = 3.9 Hz, 1H), 3.75 (s, 3H), 3.71 (s, 3H), 3.71 (s, 3H), 3.63 (s, 3H). ^13^C NMR (75 MHz, CDCl_3_): δ (ppm) = 172.6, 168.0, 153.8, 153.2, 153.2, 148.8, 147.3, 137.9, 137.0, 134.5, 132.0, 131.4, 127.5, 126.3, 126.2, 125.4, 123.6, 123.1, 121.5, 110.0, 108.3, 104.7, 104.7, 101.6, 60.8, 56.2, 52.7, 48.5, 46.7. MS (EI) (*m*/*z*) for C_29_H_25_NO_7_S [M + H]^+^ = calc. 532.13517, found 532.14245.

##### Compound **11**

For the synthesis of compound **11**, compound **7** (50 mg, 0.09 mmol) was dissolved in 5% KOH solution in methanol and stirred at room temperature for 3 days. The reaction mixture was then neutralized with 2 M HCl aq, methanol (25 mL) was added, and the mixture was evaporated to dryness. The residue was extracted with chloroform. The organic phase was dried with anhydrous MgSO_4_ and the solvent was evaporated. The product was purified with column chromatography using *n*-hexane and ethyl acetate at a ratio of 4:1 as eluent. Product **11** was obtained as a yellow solid (31.5 mg, 67% yield). ^1^H NMR (300 MHz, CDCl_3_): δ (ppm) = 7.95 (d, *J* = 7.8 Hz, 1H), 7.84 (d, *J* = 7.5 Hz, 1H), 7.44 (t, *J* = 8.3 Hz, 1H), 7.35 (t, *J* = 7.7 Hz, 1H), 7.32 (s, 1H), 6.82 (s, 1H), 6.51 (s, 2H), 6.48 (s, 1H), 5.96 (s, 1H), 5.95 (s, 1H), 4.15 (q, *J* = 6.4 Hz, 1H), 3.82 (s, 3H), 3.81 (s, 6H), 3.39 (dd, *J* = 7.8 Hz, 1H), 3.18 (dd, *J* = 8.6 Hz, 1H). ^13^C NMR (75 MHz, CDCl_3_): δ (ppm) = 168.7, 154.0, 153.4, 148.4, 146.7, 139.2, 136.9, 134.5, 134.5, 130.8, 130.5, 127.4, 126.4, 125.3, 12.3, 123.0, 121.6, 109.0, 108.1, 105.6, 101.4, 61.0, 56.2, 45.1, 33.3. ^13^C NMR (75 MHz, CDCl_3_, DEPT): δ (ppm) = 101.3, 33.2. MS (EI) (*m*/*z*) for C_27_H_23_NO_5_S [M + H]^+^ = calc. 474.13697, found 474.13788.

##### Compound **12**

To the solution of compound **7** (66.4 mg, 0.125 mmol, 1 eq) in methanol (25 mL), 3 mL of 5% KOH in MeOH solution was added and the mixture was stirred at room temperature for 2 h to form a light green precipitation. The solvent was evaporated into dryness under reduced pressure. DCM (1 mL) was added and the reaction mixture was placed in an ice bath, followed by the addition of oxalyl chloride (0.13 mmol) and DMF (1 µL) as a catalyst. The mixture was stirred for 2 h at 0–10 °C and then concentrated into dryness, followed by the addition of NH_2_NH_2_· H_2_O (0.25 mmol, 2 eq), acetic acid (1 µL), and THF (2 mL) and refluxing for 2.5 h. Water (5 mL) was added and the mixture was neutralized with HCl 2N. The mixture was extracted with EtOAc and the organic phase was dried over anhydrous Na_2_SO_4_. The product was purified by column chromatography or preparative TLC with CHCl_3_-MeOH (98:2) as eluent. Product **12** was collected as white powder with 34% yield. ^1^H NMR (300 MHz, CDCl_3_): δ (ppm) = 8.17 (s, 1H), 8.00 (ddd, *J* = 0.6, 1.2, 8.0 Hz, 1H), 7.92 (ddd, *J* = 0.6, 1.35, 7.9 Hz, 1H), 7.48 (td, *J* = 1.2, 7.7 Hz, 1H), 7.39 (td, *J* = 1.2, 7.2 Hz, 1H), 7.20 (s, 1H), 6.95 (s, 1H), 6.62 (s, 2H), 6.08 (s, 2H), 3.95 (s, 3H), 3.90 (s, 6H), 3.64 (s, 3H). ^13^C NMR (75 MHz, CDCl_3_): δ (ppm) = 169.7, 166.2, 154.2, 153.1, 149.8, 149.1, 138.3, 137.9, 135.4, 132.8, 130.9, 130.7, 129.2, 128.7, 126.8, 126.5, 125.6, 123.7, 121.7, 107.7, 107.8, 104.6, 103.7, 101.9, 61.2, 56.4, 52.4. MS (EI) (*m*/*z*) for C_29_H_23_NO_7_S [M + Na]^+^ = calc. 552.10927, found 552.10874.

##### Compound **13**

To synthesize compound **13**, compound **7** (0.06 mmol, 1 eq) was dissolved in dichloromethane (2 mL) and placed in a dry ice bath, followed by dropwise addition of DIBAL-H solution 1.0 M in dichloromethane (0.18 mmol, 3 eq). The reaction was carried out for 30 min under argon atmosphere, maintaining the temperature from −78 °C to −55 °C. Then, MeOH (3 mL) and Rochelle salt solution (10 mL) were added. The mixture was stirred for 1 h and then extracted with ethyl acetate and the organic phase was dried over anhydrous Na_2_SO_4_ and evaporated. The crude product was initially purified using column chromatography with CHCl_3_-MeOH (99:1) and then by preparative TLC in the same system. Product **13** was isolated as a yellowish powder with 8% yield. ^1^H NMR (300 MHz, CDCl_3_): δ (ppm) = 7.91 (ddd, *J* = 0.6, 1.2, 8.1 Hz, 1H), 7.86 (ddd, *J* = 0.6, 1.5, 8.1 Hz, 1H), 7.45 (td, *J* = 1.5, 7.2 Hz, 1H), 7.39 (dd, *J* = 1.5, 8.1 Hz, 1H), 7.35 (s, 1H), 6.82 (s, 1H), 6.73 (s, 1H), 6.61(s, 2H), 5.98 (d, *J* = 6.0 Hz, 1H), 4.46 (d, *J* = 6.6 Hz, 1H), 3.97 (d, *J* = 1.8 Hz, 2H), 3.87 (*s*, *3H*), 3.85 (s, 6H), 3.61 (q, *J* = 3.9 Hz, 1H). MS (EI) (*m*/*z*) for C_28_H_25_NO_6_S [M + H]^+^ = calc. 504.14745, found 504.14754.

##### Compound **14**

For the synthesis of compound **14**, compound **7** (48 mg, 0.09 mmol), methionine (56 mg, 0.4 mmol), and methanesulfonic acid (2 mL, 30 mmol) were stirred at 40◦C for 45 min. Then, the reaction mixture was neutralized with K_2_CO_3_ solution to pH 6 and extracted twice with ethyl acetate. The organic extracts were dried using anhydrous MgSO_4_, filtered, and evaporated. The product was purified with column chromatography with *n*-hexane–ethyl acetate (3:2) as eluent. Compound **14** was obtained at 11% yield. ^1^H NMR (500 MHz, CDCl_3_): δ (ppm) = 7.97 (d, *J* = 10.0 Hz, 1H), 7.85 (d, *J* = 10.0 Hz, 1H), 7.45 (t, *J* = 7.5 Hz, 2H), 7.36 (t, *J* = 10.0 Hz, 1H), 6.83 (s, 1H), 6.62 (s, 1H), 6.55 (s, 2H), 4.58 (d, *J* = 10.0 Hz, 1H), 4.39 (d, *J* = 5.0 Hz, 2H), 3.86 (s, 6H), 3.44 (s, 3H). ^13^C NMR (125 MHz, CDCl_3_): δ (ppm) = 172.6, 167.7, 153.6, 147.1, 145.3, 142.4, 134.3, 134.1, 133.6, 130.3, 130.2, 127.7, 126.3, 125.8, 125.3, 122.9, 121.4, 116.0, 115.1, 106.2,, 56.3, 56.3, 51.9, 48.9, 48.4. MS (EI) (*m*/*z*) for C_27_H_23_NO_7_S [M + H]^+^ = calc. 506.12680, found 506.12611.

##### Compound **15**

To obtained compound **15**, compound **7** (35 mg, 0.06 mmol) was dissolved in DCM (1 mL); then, TMSI (24 μL, 0.21 mmol) was added. The reaction was stirred at −4 °C for 24 h. For work up, water was added and the reaction mixture was extracted three times with DCM. The organic extracts were dried over anhydrous MgSO_4_ and evaporated. Then, column chromatography was performed with the ethyl acetate–hexane (1:1) system and preparative chromatography in the same system. The pure product in yellow solid form was obtained in 44% yield. ^1^H NMR (500 MHz, CDCl_3_): δ (ppm) = 7.96 (d, *J* = 9.0 Hz, 1H), 7.84 (d, *J* = 7.5 Hz, 1H), 7.45 (d, *J* = 1.0 Hz, 2H), 7.43 (s, 1H), 7.35 (t, *J* = 8.3 Hz, 1H), 6.87 (s, 1H), 6.61 (s, 1H), 6.56 (s, 2H), 5.98 (d, *J* = 4.5 Hz, 1H), 5.96 (d, *J* = 1.5 Hz, 1H), 5.53 (s, 1H), 4.59 (d, *J* = 8.5 Hz, 1H), 4.42 (d, *J* = 7.5 Hz, 1H), 3.87 (s, 6H), 3.43 (s, 3H). ^13^C NMR (125 MHz, CDCl_3_): δ (ppm) = 172.6, 167.3, 154.1, 148.7, 147.4, 146.7, 134.7, 133.5, 132.3, 130.4, 128.9, 127.7, 126.3, 125.4, 123.3, 121.6, 108.9, 108.8, 106.9, 105.0, 101.4, 56.6, 51.7, 49.1, 48.9 ppm. MS (EI) (*m*/*z*) for C_28_H_23_NO_7_S [M + H]^+^ = calc. 518.12680, found 518.12670.

#### 3.1.2. X-Ray Crystallography Analysis of **7**, **9** and **10**

Crystals of **7**, **9**, and **10** were examined and selected under a microscope in polarized light. Selected crystals were mounted on a κ-goniometer of an Oxford Xcalibur R diffractometer equipped with a CCD detector, and their diffraction was examined for quality control. The diffracted reflection intensities for the X-ray wavelength of 1.54184 Å were measured separately for each individually aligned single crystal. Full-matrix least-squares refinement on F^2^ was performed, and absorption corrections were applied using the analytical method [37]. The dual methods from SHELXT [38] were used to solve the structures from the diffraction data. The structure model was then refined using SHELXL-97 [39].

All hydrogen atoms, except the hydrogen atoms in some of the methyl groups, were located from a differential density map. The coordinates of the hydrogen atoms attached to the carbon atoms were treated as fixed contributors using standard geometric criteria of the idealized geometry. X-Seed [40] and Shelxle [41] were used as GUIs. They supported the refinement and visualization of the structure. X-Seed was also used to create artwork [42] in POV-RAY [43]. The final crystal and structural data for compounds **7**, **9**, and **10** are summarized in CIF format and deposited at the Cambridge Crystallographic Data Centre. All files are available free of charge from the Cambridge Structural Database at https://www.ccdc.cam.ac.uk/structures/ (CCDC 2449015–2449017)—URL accessed on 9 April 2025.

**Compound 7** was crystallized by dissolving it in warm chloroform, followed by the dropwise addition of warm n-hexane, allowing for slow crystallization. C_29_H_25_NO_7_S, Mr = 531.56, crystal size 0.459 × 0.102 × 0.050 mm^3^, orthorhombic, space group *P*2_1_2_1_2_1_, unit cell dimensions *a* = 11.4293(2) Å, *b* = 14.7160(3) Å, *c* = 32.0658(5)Å, V = 5393.26(17) Å^3^, Z = 8, Z’ = 2, α = β = γ = 90°, calculated density = 1.308 mg/cm^3^, F(000) = 2224, 121,168 reflections were collected, of which 11,541 are unique (*R_int_* = 0.0528), final *R* indices (*I* > 2σ_I_) R_1_ = 4.68%, *wR*_2_ = 12.33%, goodness of fit = 1.045, and Flack parameter = 0.004(5) confirm that the absolute configuration is correct.

**Compound 9** was crystallized by dissolving it in warm chloroform, followed by the dropwise addition of warm *n*-hexane, allowing for slow crystallization. C_29_H_24_ClNO_7_S, Mr = 566.00, crystal size 0.664 × 0.531 × 0.025 mm^3^, orthorhombic, space group *P*2_1_2_1_2_1_, unit cell dimensions *a* = 11.4586(4) Å, *b* = 14.9377(4) Å, *c* = 32.3951(13) Å, V = 5544.9(3) Å^3^, Z = 8, Z’ = 2, α = β = γ = 90°, calculated density = 1.356 mg/cm^3^, F(000) = 2352, 129,676 reflections were collected, of which 9924 are unique (*R_int_* = 0.1228), final *R* indices (*I* > 2σ_I_) *R*_1_ = 9.1%, *wR*_2_ = 26.88%, goodness of fit = 1.048, and Flack parameter = −0.002(10) confirm the proper choice of absolute structure.

**Compound 10** was crystallized by dissolving it in warm chloroform, followed by the dropwise addition of warm n-hexane, allowing for slow crystallization. C_29_H_25_NO_7_S, Mr = 531.56, crystal size 0.272 × 0.120 × 0.034 mm^3^, monoclinic, space group *P*2_1_, unit cell dimensions *a* = 13.7756(4) Å, *b* = 13.1370(4) Å, *c* = 14.2759(4) Å, V = 2512.22(13) Å^3^, Z = 8, Z’ = 2, α = γ = 90° β = 103.492(3)°, calculated density = 1.405 mg/cm^3^, F(000) = 1112, 44,748 reflections were collected, of which 10,616 are unique (*R_int_* = 0.0506), final *R* indices (*I* > 2σ_I_) *R*_1_ = 4.58%, *wR*_2_ = 11.92%, goodness of fit = 1.029, and Flack parameter = 0.005(10) confirm that the absolute configuration is correct.

### 3.2. Biological Evaluation

#### 3.2.1. Cell Culture

The synthesized compounds were tested and evaluated for their cytotoxicity using an in vitro cytotoxicity assay. We studied five cancer cell lines and one non-cancerous cell line. The cells were purchased from the American Type Culture Collection (ATCC), United States, including HeLa (cervical), LoVo (colorectal), MCF-7 (breast), SKOV3 (ovarian), B16F10 (murine melanoma), and HaCaT (human keratinocyte). Dulbecco’s modified Eagle’s medium (containing 10% FBS) was used to culture HeLa, MCF-7, B16F10, and HaCaT cells, while RPMI 1640 medium (containing 10% FBS) was used for SKOV3 and LoVo cells. The cells were cultured in 25 cm^2^ culture flasks (Corning, Corning, NY, USA) in a humidified atmosphere of 5% CO_2_ at 37 °C and passaged every three days with standard Trypsin-EDTA solution (Gibco, Thermo Fisher Scientific, Waltham, MA, USA). All compounds were initially prepared as stock solutions at a concentration of 10 mM (in DMSO). Serial dilutions were performed in the indicated medium to achieve the required concentration gradient of compounds (20 µM, 10 µM, 5 µM, 2.5 µM, 1.25 µM, 0.625 µM, 0.312 µM). The maximum percentage of DMSO used in the compound solutions was 0.1%, based on the procedure previously described by our group [44].

#### 3.2.2. The Crystal Violet Assay

This assay was performed in reference to the method already described [18]. In brief, the cell lines (HeLa, MCF-7, and SKOV-3) were seeded in the indicated medium in a 96-well culture plate (Nest Biotechnology, Wuxi, China) at a density of 1 × 10^4^ cells per well (100 µL) in quadruplicates and then incubated at 37 °C in a 5% CO_2_ atmosphere. After 24 h of incubation, 100 µL aliquots of the compounds in medium at increasing concentrations were added to each well, and the cells were further incubated for 48 h. After the incubation period, the cells were washed with PBS twice, and then with 70% ethanol, followed by staining with 0.05% crystal violet solution in water (ChemPur, Piekary Slaskie, Poland) for 20 min. The excess crystal violet was washed out with distilled water several times. The stained cells then solubilized in 1% Triton™ X-100 Surfact-Amps (POCH SA, Gliwice, Poland) or 10% PBS. The absorbance of each well was measured using Fluostar plate reader at 550 nM.

#### 3.2.3. The PrestoBlue Assay

The PrestoBlue assay was performed according to the method previously reported by Strus et al. [44]. The cells (LoVo and B16F10) were seeded in the indicated medium in a 96-well culture plate (Nest Biotechnology) at a density of 1 × 10^4^ cells per well (100 µL) and incubated at 37 °C in a 5% CO_2_ atmosphere. After 24 h of incubation, 100 µL aliquots of the compounds in medium were added to each well, and the cells were further incubated for 48 h. After 48 h, the old medium was removed, and 100 µL of fresh medium was added. Then, 10 µL of PrestoBlue solution (Thermo Fisher Scientific, Waltham, MA, USA) was added to each well and incubated for 60 min. The fluorescence intensity was measured with excitation at 544 nm and emission at 615 nm using a Fluorstar microplate reader (BMG Labtech, Ortenberg, Germany). The results were expressed as a percentage of the intensity relative to that of control cells (non-treated cells).

#### 3.2.4. Cell Cycle Analysis by Flow Cytometry

For cell cycle analysis, HeLa cells were treated for 24 or 48 h with compounds **7** and **11** at concentrations of 5 µM and 10 µM. After the incubation period, the cells were fixed in a freezing medium (ice-cold 70% ethanol solution in PBS) and stored at −20 °C. The cell suspension was then centrifuged, washed with PBS, and resuspended in PBS containing 200 µg of DNase-free RNase A (Sigma) for 20 min at room temperature. Propidium iodide (Sigma) was added to a final concentration of 5 µg/mL, and the cells were stained for at least 2 h at 4 °C. The analysis was performed using a FACSCalibur flow cytometer (Becton Dickinson, San Diego, CA, USA) and analyzed with CellQuest software, v.3.3.

#### 3.2.5. Statistical Analysis

Data analysis was carried out using GraphPad Prism 5.0.3. Calculations were performed using MS-Excel version 2021. The cytotoxic effect was expressed as the relative viability of treated cells (percentage growth control) and was calculated as follows: % relative viability = $\left(\frac{Ae-Ab}{Ac-Ab}\right)\times 100\%$, where Ae is experimental absorbance/fluorescence intensity, Ac is the absorbance/ fluorescence intensity of growth control (only cells with medium), and Ab is the absorbance/fluorescence intensity of a blank (only medium). The IC_50_ values were obtained by using the sigmoidal dose–response function in GraphPad Prism. The results were expressed as mean ± standard deviation (SD) [45].

### 3.3. In Silico Studies

The molecular docking analysis was conducted to predict and evaluate the binding interaction between compounds **7**, **11**, and **15** with tubulin. The crystal structure of tubulin (PDB ID: 1SA1) [46] was obtained from Protein Data Bank (https://www.rcsb.org/, accessed on 24 December 2024). The molecular docking calculations were performed with SwissDock via AutoDock Vina, a free web-based docking service which minimized the technical barrier to using docking software [31,32]. The ligands or compounds were inputted as SMILE notation, as previously converted by Chemical Sketch Tool (https://www.rcsb.org/chemical-sketch/, accessed on 24 December 2024), while the protein target was provided as the PDB ID. In this analysis, we specified the docking position to the podophyllotoxin binding site on β-tubulin. The docking results were then analyzed and visualized with PyMOL version 2.5.5 and Discovery Studio Visualizer 2021.

### 3.4. ADME Prediction

The SwissADME web tool (http://www.swissadme.ch/, accessed on 9 April 2025), a free website to predict the physicochemical properties of small molecules, was used to study the pharmacokinetics and drug-likeness of the most potent compounds [35]. The 2D chemical structures were input based on the molecular sketcher on ChemAxon’s Marvin JS or typing/pasting SMILE notation.

## 4. Conclusions

Based on previous results concerning the synthesis and determination of the favorable pharmacological properties of the benzothiazole derivative of podophyllotoxin (**7**), further derivatives were obtained and subjected to meticulous evaluation of their biological activity. It was confirmed that the presence of a dioxolane ring and free-rotating 3′,4′,5′-trimethoxyphenil group are important for the activity and also the stereochemistry of the methyl ester fragment with the *R* configuration, which alongside the unsubstituted benzothiazole moiety are substantial for cytotoxicity. Interestingly, compound **11**, which did not have an ester substituent at C-8′, showed a strong antimitotic effect against cancer cells via G2/M cell cycle arrest. In addition, the ADMET prediction indicated its strong GI absorption and, taking into account the fact that Lipinski’s Rule of Five is fulfilled, it seems that good prospects are opening up for the construction of new effective anticancer derivatives.

## Data Availability

Data are available on request due to restrictions, e.g., privacy or ethical considerations. The data presented in this study are available on request from the corresponding authors.

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
