# Peer review of "Synthesis of Novel Podophyllotoxin–Benzothiazole Congeners and Their Biological Evaluation as Anticancer Agents"

_ijms, 2025, doi:10.3390/ijms26136033_

Round 1

Reviewer 1 Report

Comments and Suggestions for Authors

Rai’dah et al. on “Synthesis of novel podophyllotoxin-benzothiazole congeners and their biological evaluation as anti-cancer agents.

The topic is relevant and current in light of the therapeutic challenges facing the treatment of cancers. The chemistry is adequate and detailed to enable reproducibility. The characterization data confirms the structures of the synthesized derivatives. The pharmacological methodology design and methods employed are appropriate. The results are adequately reported and thoroughly discussed. The manuscript is coherently written and read quite well. However, several issues must be addressed prior to its consideration for acceptance.

Introduction

The introduction is extensive on the rationale for the derivatization podophyllotoxin which is commendable. However, there is a lack of information on cancer statistics in general and specifically on the choice of cancer cell lines of interest. In other words, what is the motivation for targeting these cancer-types? The justification should be supported by latest statistics of cervix, colorectal, breast, ovarian and melanoma cancers.

Pharmacology

HaCaT cell line is used in the study to assess the basal toxicity of the synthesized compounds, some of which namely 7 and 11 are significantly toxic to it, the precursor podophyllotoxin 1 being extremely toxic as well. Hence, the observed activity of the two derivatives may not intrinsic as their inmate cytotoxicity may be a contributor to this apparent. As such the selective indexes (SI) should be calculated for each active compound against each cell line with SI compound = IC50 (HaCaT)/IC50 (cancer cell).

A potential cancer hit is regarded as having an IC50 <10 μM and selectivity index >10 in terms of synthesized compounds, which matches the criteria for antileishmanial hit compounds (Katsuno et al., 2015, Nature reviews drug discovery, 14(11):751-758).

Thus, Table 1 must be revised and the biological activity discussion amended accordingly to identify whether the study uncovered any anticancer hit.

On these grounds, I conditionally support acceptance of this manuscript for publication on IJMS. A revised version must be submitted. 

Author Response

We would like to thank the reviewers for all their comments and the effort put into reviewing the manuscript. We have addressed all the remarks and made the appropriate revisions. The changes have been highlighted in yellow in the revised manuscript.

Comment 1: Introduction

The introduction is extensive on the rationale for the derivatization podophyllotoxin which is commendable. However, there is a lack of information on cancer statistics in general and specifically on the choice of cancer cell lines of interest. In other words, what is the motivation for targeting these cancer-types? The justification should be supported by latest statistics of cervix, colorectal, breast, ovarian and melanoma cancers.

Response to comment 1:

The justification and the statistics were included in the “Introduction” (page 1, ref.7-9). Moreover, in the results and discussion part “Biological activity” part an adequate note was added (page 5).

Comment 2: Pharmacology

HaCaT cell line is used in the study to assess the basal toxicity of the synthesized compounds, some of which namely 7 and 11 are significantly toxic to it, the precursor podophyllotoxin 1 being extremely toxic as well. Hence, the observed activity of the two derivatives may not intrinsic as their inmate cytotoxicity may be a contributor to this apparent. As such the selective indexes (SI) should be calculated for each active compound against each cell line with

 SI compound = IC50 (HaCaT)/IC50 (cancer cell).

A potential cancer hit is regarded as having an IC50 <10 μM and selectivity index >10 in terms of synthesized compounds, which matches the criteria for antileishmanial hit compounds (Katsuno et al., 2015, Nature reviews drug discovery, 14(11):751-758).

Thus, Table 1 must be revised and the biological activity discussion amended accordingly to identify whether the study uncovered any anticancer hit.

Response to comment 2: 

We have revised Table 1 to include the selectivity index (SI) values for a more detailed evaluation of each compound’s selectivity profile. Furthermore, the discussion on biological activity has been updated to reflect these changes and to assess whether any of the synthesized compounds meet the hit criteria for anticancer agents. Based on the revised data, we highlight compounds 9 and 15 as the most promising candidates, exhibiting selective cytotoxicity toward specific cancer cell lines while sparing non-tumorigenic HaCaT cells (page 7, ref. 28).

Reviewer 2 Report

Comments and Suggestions for Authors

Czarnocki and his co-workers have reported “Synthesis of novel podophyllotoxin-benzothiazole congeners and their biological evaluation as anti-cancer agents. The authors synthesized a series of novel podophyllotoxin derivatives incorporating benzothiazole scaffolds and evaluated their in vitro cytotoxic activity against five cancer cell lines. Among these, two compounds exhibited the most potent cytotoxic effects across all tested cell lines. Further mechanistic studies revealed that these compounds inhibit cancer cell proliferation by inducing G2/M phase arrest in HeLa cells. However, several revisions are necessary before this manuscript can be considered for publication in the International Journal of Molecular Sciences.

Comments:

  1. While the authors performed molecular docking studies to explore the binding mode of the compounds with β-tubulin, molecular dynamics (MD) simulations should also be conducted to gain deeper insights into the stability and detailed interactions of the ligand–protein complexes at the molecular level.
  2. To validate the docking results, biochemical assays targeting β-tubulin are recommended.
  3. The NMR spectral data are not presented in the correct format and should be revised accordingly.
  4. The manuscript contains numerous typographical errors, which must be carefully corrected.
Comments on the Quality of English Language

 The English could be improved to more clearly express the research.

Author Response

We would like to thank the reviewers for all their comments and the effort put into reviewing the manuscript. We have addressed all the remarks and made the appropriate revisions. The changes have been highlighted in yellow in the revised manuscript.

Comment 1:

While the authors performed molecular docking studies to explore the binding mode of the compounds with β-tubulin, molecular dynamics (MD) simulations should also be conducted to gain deeper insights into the stability and detailed interactions of the ligand–protein complexes at the molecular level.

Response to comment 1:

Molecular dynamics (MD) simulations, while valuable, in our opinion are not essential for the initial assessment of drug–protein interactions. In early-stage drug discovery, molecular docking is typically sufficient to predict binding poses, estimate binding affinities, and prioritize compounds for further analysis. MD is more appropriate for later stages, where it helps to evaluate the stability of ligand–protein complexes, conformational flexibility, and dynamic behavior under physiological conditions. Therefore, MD serves as a complementary tool for refinement and validation, rather than a prerequisite for initial screening. At this stage we were unable to perform the detailed MD analysis.

Comment 2:

To validate the docking results, biochemical assays targeting β-tubulin are recommended.

Response to comment 2:

We fully agree that biochemical assays targeting β-tubulin would provide important experimental validation of the docking results. However, due to limitations within the submission timeline, we are unable to perform such assays at this stage. We consider this an important direction for future research and plan to include β-tubulin-targeted biochemical studies in our follow-up investigations to further confirm the molecular mechanism of action of the most promising compounds.

Comment 3:

The NMR spectral data are not presented in the correct format and should be revised accordingly.

Response to comment 3:

The format of the NMR spectral data has been adjusted to match the format used in previous manuscripts published in Int. J. Mol. Sci. (https://doi.org/10.3390/ijms25126421).

Comment 4:

The manuscript contains numerous typographical errors, which must be carefully corrected.

Response to comment 4:

The manuscript was checked carefully and corrected.

Round 2

Reviewer 1 Report

Comments and Suggestions for Authors

The authors have adequately addressed all the shortcomings that arose from the initial version of the manuscript. Hence, I support the acceptance and publication of this revised version as is. 

Reviewer 2 Report

Comments and Suggestions for Authors

The authors have responded to all the comments I made earlier.